# Efficacy of Topically Administered Dihydroartemisinin in Treating Papillomavirus-Induced Anogenital Dysplasia in Preclinical Mouse Models

**DOI:** 10.3390/v14081632

**Published:** 2022-07-26

**Authors:** Laura C. Gunder, Simon Blaine-Sauer, Hillary R. Johnson, Myeong-Kyun Shin, Andrew S. Auyeung, Wei Zhang, Glen E. Leverson, Ella T. Ward-Shaw, Renee E. King, Stephanie M. McGregor, Kristina A. Matkowskyj, Paul F. Lambert, Evie H. Carchman

**Affiliations:** 1Department of Surgery, School of Medicine and Public Health, University of Wisconsin, 600 Highland Ave, Madison, WI 53792, USA; gunderl@surgery.wisc.edu (L.C.G.); hjohnson5@uwhealth.org (H.R.J.); aauyeung@kent.edu (A.S.A.); leverson@surgery.wisc.edu (G.E.L.); 2McArdle Laboratory for Cancer Research, School of Medicine and Public Health, University of Wisconsin, 1111 Highland Ave, Madison, WI 53705, USA; blainesauer@wisc.edu (S.B.-S.); myeongkyun.shin@wisc.edu (M.-K.S.); etward@wisc.edu (E.T.W.-S.); kingr@surgery.wisc.edu (R.E.K.); lambert@oncology.wisc.edu (P.F.L.); 3Department of Pathology and Laboratory Medicine, School of Medicine and Public Health, University of Wisconsin, 3170 UW Medical Foundation Centennial Building (MFCB), 1685 Highland Avenue, Madison, WI 53705, USA; wzhang566@wisc.edu (W.Z.); smcgregor@uwhealth.org (S.M.M.); matkowskyj@wisc.edu (K.A.M.); 4University of Wisconsin Carbone Cancer Center, School of Medicine and Public Health, University of Wisconsin, 600 Highland Ave, Madison, WI 53705, USA; 5William S. Middleton Memorial Veterans Hospital, 2500 Overlook Terrace, Madison, WI 53705, USA

**Keywords:** dihydroartemisinin, DHA, artemisinin, papillomavirus, HPV, MmuPV1, anal dysplasia, cervical dysplasia, cancer, preclinical model

## Abstract

The artemisinin family of compounds is cytopathic in certain cancer cell lines that are positive for human papillomaviruses (HPV) and can potentially drive the regression of dysplastic lesions. We evaluated the efficacy of topical dihydroartemisinin (DHA) on cervical dysplasia and anal dysplasia in two papillomavirus mouse models: *K14E6/E7* transgenic mice, which express HPV16 oncogenes; and immunodeficient NOD/SCID gamma (NSG) mice infected with *Mus musculus* papillomavirus (MmuPV1). Mice started treatment with DHA at 25 weeks of age (*K14E6/E7*) or 20 weeks post infection (MmuPV1-infected), when the majority of mice are known to have papillomavirus-induced low- to high-grade dysplasia. Mice were treated with or without topical DHA at the cervix or anus and with or without topical treatment with the chemical carcinogen 7,12 dimethylbenz(a)anthracene (DMBA) at the anus of in transgenic mice to induce neoplastic progression. Mice were monitored for overt tumor growth, and tissue was harvested after 20 weeks of treatment and scored for severity of histological disease. For MmuPV1-infected mice, anogenital lavages were taken to monitor for viral clearance. Tissues were also evaluated for viral gene expression at the RNA and/or protein levels. Treatment with topical DHA did not reduce dysplasia in the anogenital tract in either papillomavirus-induced mouse model and did not prevent progression to anal cancer in the DMBA-treated *K14E6/E7* mice.

## 1. Introduction

Human papillomavirus (HPV) infection is associated with more than 95% of anogenital dysplasia cases [1]. Persistent HPV infection causes transformation of normal epithelial cells and can lead to precancerous changes in cell architecture called dysplasia. Dysplasia can progress from low-grade intraepithelial lesions (LSILs) to high-grade intraepithelial lesions (HSILs) and, ultimately, to squamous cell carcinoma (SCC). This process of neoplastic progression is thought to be similar in both cervical and anal cancers. Treatment of HSIL is effective in reducing the incidence of SCC [2,3]. However, there are currently no FDA-approved topical treatments for anogenital dysplasia to prevent progression to cancer. Furthermore, the incidence of anal cancer continues to increase by almost 3% annually despite the availability of a prophylactic HPV vaccine. Current standard-of-care treatments for anogenital dysplasia can have significant adverse effects on patient quality of life, including pain, skin irritation, and ulceration [4]. In addition to poor patient tolerance of these treatments, current anogenital dysplasia treatment regimens are associated with recurrence rates greater than 10–50% [5,6]. Therefore, new treatments are needed to prevent the development of HPV-associated cancers.

In this study, we investigate whether topical application of dihydroartemisinin (DHA) decreased papillomavirus-induced neoplasia in two mouse models of papilloma-associated carcinogenesis of the anogenital tract. These preclinical models include transgenic *K14E6/E7* mice that spontaneously develop anogenital dysplasia and a model of neoplastic disease mediated by infection of the desired area (cervix or anus) with *Mus musculus* papillomavirus (MmuPV1). *K14E6/E7* mice constitutively express the high-risk HPV16 oncoproteins E6 and E7 in their epithelia. Without carcinogen treatment, these mice develop progressive anal disease, from normal to LSIL to HSIL to SCC, similar to the progression in humans [7]. Mouse papillomavirus (MmuPV1) was discovered in 2011 [8] and has become an invaluable tool for studying papillomavirus-associated carcinogenesis in vivo. Unlike the *K14E6/E7* model, this represents the first mouse model for translational studies of anti-papillomaviral and anti-anogenital cancer therapeutic agents in the context of a natural viral infection.

Artemisinin, derived from the *Artemisia annua* plant, has long been used in traditional Chinese medicine. Together with its semisynthetic derivatives, artesunate and dihydroartemisinin (DHA), artemisinin is an effective antimalarial drug [9,10]. In recent years, artemisinin and its analogs have been highlighted for their potential use in treating a variety of other diseases, including cancer [11,12]. The anticancer activities of artemisinin family compounds have been observed in both in vitro and in vivo models [9,13,14]. Specifically, DHA has shown promise in models of cervical dysplasia, including human papillomavirus (HPV)-transformed cervix cell lines and human cervical cancer cells [15]. DHA is an ideal treatment for anogenital dysplasia because it is hydrophobic and shelf-stable, lending itself well to topical application. Topical application would be a preferred treatment route in the anal and female reproductive tracts because it facilitates patient self-application and allows for targeted delivery only to affected tissues, thereby avoiding variations in drug metabolism, given the high first-pass effect of artesunate compounds associated with the enteral route [16]. Additionally, topical DHA can be delivered in a suppository formulation for anogenital treatment, as was demonstrated in a cervical dysplasia study [17]. The safety profile of suppository delivery is well-documented and frequently used in children for the treatment of malaria [18].

## 2. Materials and Methods

### 2.1. Mice

Two strains of mice were used in this study: *K14E6/E7* transgenic mice (9–12 mice/group) and NOD/SCID gamma (NSG) mice (for anal experiments, 6–8 mice per group; for cervical experiments, 3–4 mice per group). The *K14E6/E7* transgenic mice were generated by crossing two separate strains of mice expressing either the HPV16 E6 [19] or E7 oncogene [20] driven from a K14 promoter, as described by Stelzer et al. (2010) [21]. The NOD/SCID gamma (NSG) mice were purchased from Jackson Laboratory (stock number: 005557) and bred by the UW-Madison Biomedical Research Models Services Laboratory. All mice were housed in American Association for Accreditation of Laboratory Animal Care-approved facilities: McArdle Laboratory Cancer Center Animal Care Facility or Wisconsin Institute for Medical Research (WIMR) Animal Care Facility. These studies were performed in accordance with approved Institutional Animal Care under approved animal protocols M006333 and M005871. Mice were monitored weekly for local side effects. Mice were treated for a 20-week period or until they met euthanasia requirements of the animal protocol.

### 2.2. 7,12 Dimethylbenz[a]anthracene (DMBA) Treatment of K14E6/E7 Transgenic Mice

Male and female *K14E6/E7* transgenic mice were treated topically at the anus once a week with 0.12 μmoles of 7,12 dimethylbenz[a]anthracene ((DMBA), Cat. No. D3254, Sigma Aldrich, Saint Louis, MO, USA) solution dissolved in 60% acetone/40% dimethylsulfoxide ((DMSO), Cat. No. BP231-1, Thermo Fisher Scientific, Waltham, MA, USA) at least 30 min prior to any other topical treatment to allow for proper absorption of the drug into the tissue. Mice began treatment at 25 weeks of age, when the majority of these transgenic mice spontaneously develop HSIL of the anus.

### 2.3. Infection with MmuPV1

Male and female NSG mice were infected in the anus at 8–10 weeks of age using a Greer pick (Cat. No. GP-1C, Greer Laboratories, Inc., Lenoir, NC, USA) dipped in a 3 × 10^8^ viral genome equivalents (VGE)/µL MmuPV1 stock, as previously described [22], which leads to inoculation with approximately 5 × 10^8^ VGE. Female NSG mouse reproductive tracts were infected as described by Spurgeon et al. (2019) [23]. Female mice were infected in the cervicovaginal tract at 24–26 weeks of age.

### 2.4. Dihydroartemisinin (DHA) Treatment of Mice

DHA treatment concentration for all mice was adapted from Disbrow et al. (2005) [12]. *K14E6/E7* transgenic mice began treatment at 25 weeks of age, the age at which more than 75% of these mice spontaneously develop HSIL of the anus. Mice were treated topically at the anus five days a week (Monday–Friday) with 20 µL of a solution containing 78 mM DHA (dihydroartemisinin, Item No. 19846, Cayman Chemical Company, Ann Arbor, MI, USA) in 100% DMSO or vehicle-only control.

For NSG mice with MmuPV1-infected anuses, beginning 20 weeks post infection, when the majority of mice had previously developed HSIL [22], mice were treated five days a week (Monday–Friday) topically at the anus with 20 µL of a solution of 78 mM DHA in 100% DMSO for 20 weeks. NSG mice infected in the cervicovaginal tract were treated five days a week (Monday–Friday) with 20 µL of a solution of 78 mM DHA in DMSO beginning 26 weeks post infection. Mice infected in the cervix with MmuPV1 concluded treatment and were sacrificed after 14 weeks of DHA treatment due to poor health and advanced age.

### 2.5. Lavage and qPCR for MmuPV1 DNA

Immediately prior to sacrifice, NSG mice anogenital tracts were lavaged with 25 µL of phosphate-buffered saline (PBS), triturating 3–5 times in the anal or vaginal tract before collection. Lavages were frozen at −20 °C, and DNA was subsequently extracted using a DNeasy^®^ Blood and Tissue Kit (Ref No. 69581; Qiagen, Hilden, Germany) according to the manufacturer’s protocol. qPCR for MmuPV1 DNA or a mouse genomic DNA housekeeping gene (18S) was performed in 10 µL reactions using a method and primers described by Hu et al. [24].

### 2.6. Mouse Tissue Collection

After completing the treatment course, mice were sacrificed, and anogenital tissue was collected. The tissues were fixed in 4% paraformaldehyde for 24 h and then placed in 70% ethanol. After fixation, the tissues were processed, embedded in paraffin, and serially sectioned with 5 μm thickness.

### 2.7. Histology

Sections were stained with hematoxylin and eosin by the University of Wisconsin Carbone Cancer Center (UWCCC) Experimental Pathology Laboratory or by E.W.S. Sections were evaluated by trained pathologists blinded to treatment groups for evidence of anogenital dysplasia or carcinoma. Anal sections were scored by trained gastrointestinal pathologists (W.Z., *K14E6/E7* anal tissue; K.A.M., MmuPV1-infected anal tissue). MmuPV1-infected female reproductive tract sections were scored by a trained gynecological pathologist (S.M.M.). Low-grade dysplasia and high-grade dysplasia are used interchangeably with LSIL and HSIL, respectively.

### 2.8. Immunohistochemistry for K14E6/E7 Samples

Expression of HPV E6 and E7 proteins was assessed by immunohistochemistry (IHC). Paraffin-embedded sections mounted on glass slides were deparaffinized in xylene, rehydrated in a series of ethanol washes, subjected to citrate-buffered antigen retrieval, DNA denatured with 2N HCl, and blocked with 5% horse serum. Sections were then stained with either human papillomavirus type 16/18 E6 antibody (C1P5) (Cat No. GTX60410, GeneTex, Inc., Irvine, CA, USA; 1:250 in 5% horse serum with phosphate-buffered saline (PBS)) or human papillomavirus type 16 E7 antibody [6F3] (Cat No. GTX20070, GeneTex, Inc., Irvine, CA, USA; 1:500 in 5% horse serum with phosphate-buffered saline (PBS)) overnight at 4 °C in a humidified chamber. Each slide was then washed with 1× PBS and incubated with VectaStain universal secondary antibody (Cat No. BP-1400-50, Vector Laboratories, Burlingame, CA, USA, 1:50 in 5% horse serum in PBS;), followed by VectaStain Elite ABC reagent (R.T.U.) (Cat No. PK-7100, Vector Laboratories, Burlingame, CA, USA) as per the manufacturer’s protocol. Finally, sections were incubated with 3,3′-diaminobenzidine substrate (DAB Peroxidase (HRP) Substrate; Cat No. SK-4100, Vector Laboratories, Burlingame, CA, USA), counterstained with hematoxylin, dehydrated, and mounted. IHC was performed at the same time for all samples and for the same duration. Light microscopy was performed, and images were acquired at 200× magnification using a Zeiss Axio Imager M2 imaging system. Images were then analyzed with ImageJ version 2.0.0 (Fiji distribution). Images underwent color deconvolution using the H DAB vector. The area of interest was selected for color 2. The image was then thresholded and calibrated for optical density (OD) function for analysis. Given the small area of interest, only one field was analyzed per sample.

### 2.9. MmuPV1-E6/E7 RNA ISH

MmuPV1 viral transcripts were detected using RNAscope 2.5 HD Assay-Brown (Cat No. 322300, Advanced Cell Diagnostics, Newark, CA, USA) according to the manufacturer’s instructions [25] with probes specific to the MmuPV1 E6/E7 region (Cat No. 473281, Advanced Cell Diagnostics, Newark, CA, USA). Tissue sections were treated following protease treatment and prior to probe hybridization with 20 units of DNase I (Cat No. EN0521, Thermo Fisher Scientific, Waltham, MA, USA) for 30 min at 40 °C. Slides were counterstained with hematoxylin before mounting and cover slipping. Light microscopy was performed, and images were acquired at 200× magnification using a Zeiss Axio Imager M2 imaging system. Images were then analyzed with ImageJ version 2.0.0 (Fiji distribution). Images underwent color deconvolution using the H DAB vector. The area of interest was selected for color 2. The image was then thresholded and calibrated for optical density (OD) function for analysis. Given the small area of interest, only one field was analyzed per sample.

### 2.10. Dual L1/K14 TSA Immunofluorescence

Dual L1/K14 tyramide signal amplification (TSA) immunofluorescence was performed as previously described on samples from MmuPV1-infected mice [26]. This protocol utilizes L1 rabbit pAb (1:5000), which was kindly provided by Dr. Chris Buck (National Institutes of Health), and K14 rabbit pAb (1:1000) (Cat No. 905301; BioLegend, San Diego, CA, USA). Slides were imaged using a Zeiss Axio Imager M2 imaging system. Each 200× image was imported into Image J version 2.0.0 (Fiji distribution) and underwent the following processing. Images were split into three channels (488, 594, and Hoescht). All images were thresholded using the default dark background. The area of interest, the anorectal transition zone, was manually selected, and RawIntDen (raw integrated density) values were measured for the region of interest to determine the intensity of the fluorescent signal. The RawIntDen was then normalized for the area of the selected region (RawIntDen/Area).

### 2.11. Statistical Analysis

To detect at least a 50% difference in tumor incidence with a type I error rate of 5% and a type II error rate of 20% (80% power) between groups, 12 mice per group were needed. Overall type I error rates were controlled by omnibus test for an association between the factor and the outcome of interest. Additional pairwise comparisons were unadjusted. Comparison of *K14E6/E7* survival curves were analyzed via log-rank tests (Mantel-Cox): control versus DHA only and DMBA only versus DHA + DMBA. Chi-square tests or Fisher’s exact tests were used to examine differences between the final histology grade of mouse treatment group after the treatment period. *K14E6/E7* group analyses were performed using SAS statistical software version 9.4 (SAS Institute Inc., Cary, NC, USA). Unpaired *t*-tests were utilized for immunohistochemical staining for E6 and E7 and RNAscope ISH for MmuPV1-E6-E7. Final viral loads as assessed by qPCR for both anal and cervicovaginal groups were analyzed using Mann–Whitney tests. Data analysis for NSG mouse groups was performed using Graphpad Prism Version 9.3.1(San Diego, CA, USA) (350). Statistical significance was defined as a *p*-value of 0.05 or less. Significance was assigned as * *p* < 0.05, ** *p* < 0.01, *** *p* < 0.001.

## 3. Results

### 3.1. Topical DHA Treatment Did Not Reduce Severity of Anal Neoplastic Disease or Tumor-Free Survial in a K14E6/E7 Transgenic Mouse Model

To assess whether DHA is able to cause regression of anal dysplasia and/or prevent neoplastic progression to anal carcinoma induced by HPV16 oncogenes E6 and E7 [21], we allowed these mice to age to week 25, as by this time, these mice have spontaneously developed high-grade anal dysplasia prior to treating one group with DHA alone or vehicle alone to assess whether DHA caused regression of anal dysplasia [7]. No overt tumors were observed in either of these two groups of mice over the 20-week treatment period (Figure 1A).

To determine whether DHA prevents progression of dysplasia to cancer, we started at the same time point (25 weeks of age), treating two cohorts of mice weekly with DMBA for 20 weeks; one cohort was also given DHA, whereas the other was not. DMBA-treated *K14E6/E7* mice develop overt tumors that can be visually observed [21]. We did not observe a difference in tumor-free survival between mice treated with DHA and DMBA vs. those treated with DMBA only (*p*-value = 0.8112) (Figure 1A).

Blinded histopathological analysis was performed by a trained gastrointestinal pathologist on all anal tissue samples to determine the most severe grade of neoplastic disease in each anal canal. Representative H&E-stained sections are shown (Figure 1B). First, to assess whether DHA caused regression of pre-existing anal dysplasia, we compared the DHA-alone group to the vehicle-only control group of *K14E6/E7* transgenic mice. We observed no statistically significant difference in final anal histology (*p*-value = 0.4237), indicating that DHA does not induce regression of anal dysplasia. Second, to assess whether DHA prevented progression of anal dysplasia to cancer, we compared the DMBA-only-treated group to the DHA + DMBA-treated group. There was no statistically significant difference in severity of disease, and mice in both groups progressed to SCC (*p*-value = 0.5901) (Table 1).

We next evaluated whether DHA treatment had any effect on the level of expression of HPV16 E6 and E7 at the protein level by IHC (Figure 1B,C). In our *K14E6/E7* transgenic mice, there was no statistically significant difference in E6 expression between vehicle-only and DHA-only tissue (*p*-value = 0.9889). There was a statistically significant reduction in mean E6 protein expression in the DMBA-only group compared to the DHA + DMBA group (*p*-value = 0.0003). There was no statistically significant difference between E7 expression for either set of treatment groups when compared to their respective controls: no treatment vs. DHA-only tissue (*p*-value = 0.2171) and DMBA vs. DHA + DMBA group (*p*-value = 0.2572) (Figure 1C).

### 3.2. Topical DHA Treatment Did Not Reduce MmuPV1-Mediated Anal Dysplasia

We next assessed whether DHA affected anal dysplasia induced by infection of NSG mice with MmuPV1. Mice infected with MmuPV1 were treated with vehicle or DHA for a 20-week period beginning at 20 weeks post infection, which is a time when we previously documented MmuPV1-infected mice to have developed high-grade anal dysplasia [14]. At the endpoint, pathological analysis indicated that all mice retained HSIL, regardless of the treatment group (Table 2). Representative images of HSILs of vehicle-treated and DHA-treated mice are shown in Figure 2A. There also was no significant difference in the viral load present in anal lavages from NSG mice infected with MmuPV1 treated with vehicle as compared to DHA-treated counterparts (*p*-value = 0.4136) (Figure 2B). There was a ~ twofold reduction in viral RNA in MmuPV1-infected anal tissue of NSG mice treated topically with DHA as compared to the vehicle-treated cohort (*p*-value = 0.0454) based on measurement of the signal from MmuPV1-specific RNAscope by optical density (Figure 2C).

### 3.3. Topical DHA Treatment Did Not Reduce MmuPV1-Mediated Cervical Dysplasia

We also assessed whether DHA had any effect on cervicovaginal dysplasia induced by MmuPV1 infection of female reproductive tracts. Treatment was as described above for the mice infected at the anus. At the endpoint, pathological analysis indicated that all mice had LSIL of the cervicovaginal tract, regardless of treatment (Table 3). The endpoint for the analysis in the female mice infected with MmuPV1 in their cervix/vagina was earlier than for other groups (14 weeks of treatment) because in this group of mice, dyspnea was observed in a subset of mice necessitating euthanasia. Representative images are presented in Figure 3A. There was no significant difference in the amount of viral DNA from cervical lavages between MmuPV1-infected NSG mice treated with vehicle only and DHA-treated counterparts (*p*-value = 0.5374) (Figure 3B). There was also no significant difference in viral RNA between vehicle-treated and DHA-treated *NSG* mice based on RNAscope analysis (*p* = 0.7144) (Figure 3A,C).

## 4. Discussion

HPV-associated anogenital cancers are a major clinical problem worldwide. There is a clinical need for novel therapeutics targeting the infected cells of precancerous lesions to prevent development of cancer [4]. Current treatments for anogenital cancers are not targeted to HPV-infected cells and have high recurrence rates, particularly for anal disease, resulting in the continual use of treatments and considerable healthcare costs [27,28,29].

Artesunate, another artemisinin derivative similar to DHA, has a well-documented toxicity profile with respect to treating patients with malaria, for which it has been shown to be well-tolerated when administered intrarectally as a suppository [30,31,32]. Artesunate has been shown to be so safe, and the World Health Organization (WHO) has approved it as first-line therapy for acute malaria in children. Worldwide, approximately two million people have been treated with artesunate, providing a wealth of safety data. Epithelial cells that are infected with HPV overexpress the transferrin receptor and have increased levels of intracellular iron compared to normal cells [33]. Artesunate contains an endoperoxide bride that reacts with intracellular ferrous iron to generate free radicals. Reactive oxygen species mediate cell death through a variety of mechanisms, including apoptosis, necrosis, and a unique caspase-independent pathway known as ferroptosis [34,35]. Furthermore, iron metabolism is altered in cancers, with higher intracellular iron levels found in malignant cells compared to normal cells [36,37,38]. Disbrow et al. showed that cells expressing the HPV16 oncoproteins E6 and E7 exhibit differential cytotoxicity to DHA and that this effect is dependent on the iron-mediated formation of reactive oxygen species (ROS) [15]. Furthermore, they found that DHA was able to inhibit tumor formation in an oral canine papillomavirus animal model. These differences between HPV-infected cells and normal cells potentially allow artesunate to preferentially target diseased tissues. Preliminary data from a recent clinical trial (NCT02354534) of intravaginal artesunate suppositories to treat cervical dysplasia demonstrated 68% histologic regression and 48% HPV clearance with mild mucosal irritation [17].

The use of DHA to treat anal dysplasia has not been previously explored using in vivo models. In this study, we evaluated the use of topical DHA, a semisynthetic derivative of artemisinin, in two mouse models of papillomavirus-associated anal disease. The first model we used was a *K14E6/E7* transgenic mouse model, in which the mice are bred to constitutively express the HPV16 E6 and E7 oncogenes. By 25 weeks of age, the majority of mice develop high-grade dysplasia of the anus, and when challenged with DMBA for a further 20 weeks, the majority develop anal cancer. DHA treatment failed to prevent tumor development in mice treated with DHA + DMBA, with comparable tumor-free survival rates to those treated solely with DMBA. This result is consistent with histologic evidence of cancer, in which mice given DHA + DMBA had a similar rate of histologically proven anal cancer to those of mice treated with DMBA only. The DHA-treated group appeared to have slightly worse outcomes, although the difference was not statistically significant. DHA + DMBA-treated mice had slightly higher rates of anal cancer, and not a single DHA-only mouse had normal histology. This slight variation in outcome may be due to the increased irritation of anal tissue from five-times weekly topical drug application with DHA as compared DMBA-only mice, which only received once-weekly anal application of drug and no other topical treatments. Given the lack of effectiveness in reducing anal cancer, we were not surprised to find that E6 and E7 protein expression did not significantly differ for most treatment groups.

Our second mouse model of anal dysplasia, MmuPV1 infection of the anal tract of NSG mice, was utilized to highlight the role DHA could play in viral clearance. When tissue was analyzed for histology, there were no changes in histological outcome between treatment groups, as all DHA-treated and vehicle-treated mice developed HSIL of the anus (Figure 2). This outcome is consistent with previously published work [22]. The lack of significant findings between treatment groups was consistent with our *K14E6/E7* transgenic mouse data (Figure 1). There was no change in viral load between groups as assessed by anal lavage followed by qPCR for MmuPV1 DNA. However, we did observe a twofold decrease in viral RNA in DHA-treated NSG mice around the area of the anal transition zone, where MmuPV1 has previously been shown to preferentially localize [22]. Therefore, there may be some diminution in viral expression with DHA treatment. There is also a potential for this to demonstrate integration of the virus, although this can be neither confirmed nor denied with the above experiments. Prior data have shown that artemisinin is effective in treating cervical dysplasia [17,39]. Given the dissimilarities between these previously published human cervical data and our anal mouse model findings (Figure 1 and Figure 2; Table 1 and Table 2), we tested our topical DHA solution in our viral *NSG* mouse model of the cervicovaginal tract. We were unable to recapitulate the positive preventative effects of DHA in this model (Figure 3, Table 3). This viral infection model was adapted from a study in which *FoxN1nu/nu* female reproductive tracts were infected with MmuPV1 and developed either high-grade dysplasia or cancer 120 days post infection [23]. Inconsistencies in histologic outcome could be due to differences in mouse strains, as well as time of initial infection and post-infection sacrifice. Comparable to the NSG anal model, there were no differences in viral load between the two groups of mice. Unlike the anal model, there were no differences observed between viral RNA levels.

The lack of response to topical DHA could have occurred for several reasons. First, DHA is not effective in the treatment of anogenital dysplasia. Second, the anticancer and antiviral effects of DHA may work through a mechanism that is not captured in our mouse models of disease. Experiments were not powered to detect differences <50% between experimental and control groups. The dose of drug or vehicle chosen may not be sufficient to deliver a therapeutic dose. Failure to observe a response in the cervix, in contrast to the success observed in cervical clinical trials, may be related to the ability of DHA to activate the adaptive immune response, which is not present in our immunodeficient MmPV1 mouse model. Finally, we did not assess the potential benefits of DHA as part of a combination therapy.

Limitations for this study include a lack of drug quantification to evaluate proper absorption into the tissues of interest. In future studies with topical application of drug, serum samples and tissue from all treatment groups will be analyzed by high-performance liquid chromatography (HPLC). Drug concentrations will be measured and compared to those of clinically relevant concentrations in patients. Secondly, we did not know the exact histology of the anogenital tissue prior to treatment; given the small area of interest, it is not possible to biopsy the target tissue prior to treatment. In an attempt to overcome this issue, we used previous time-course data in both models to determine the time point when greater than 75% of mice have dysplastic lesions.

## 5. Conclusions

In conclusion, we did not observe an antipathogenic or antiviral effects of topical DHA treatment in our mouse models of anogenital neoplastic disease.

## Figures and Tables

**Figure 1 viruses-14-01632-f001:**
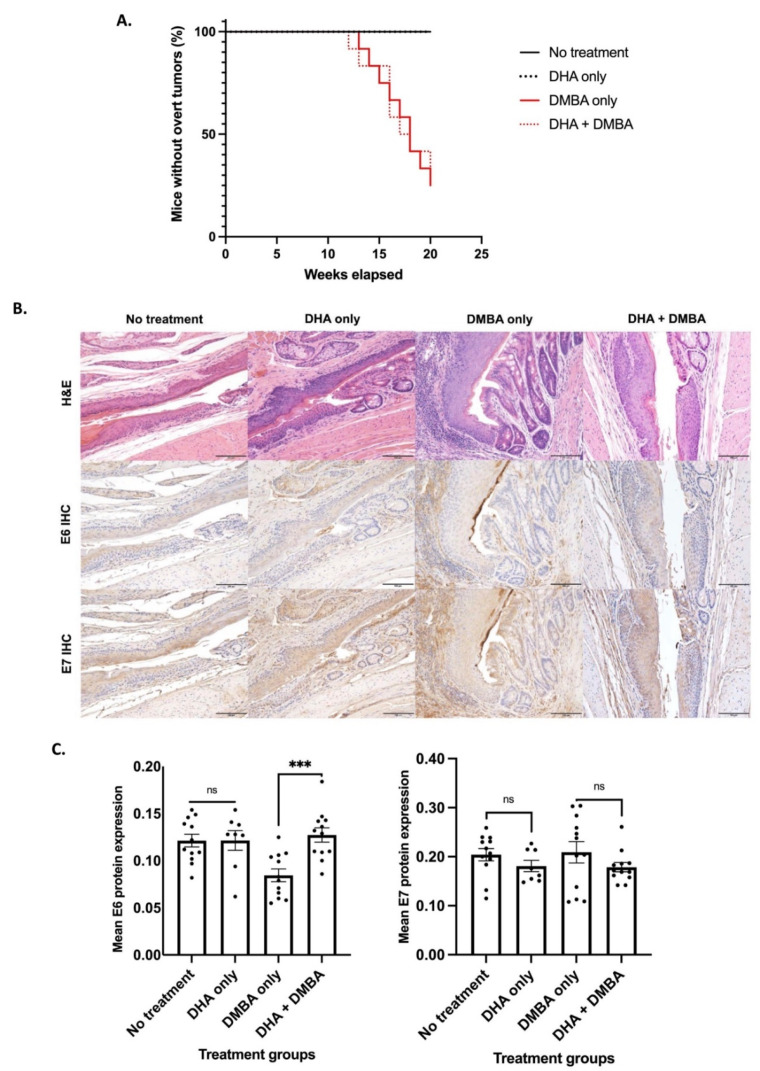
*K14E6/E7* mice began treatment at 25 weeks of age and were either left untreated (N = 12), treated with topical DHA (N = 9), treated once weekly with topical DMBA (N = 12), or given topical DHA in conjunction with DMBA (N = 12). (**A**) DHA-treated mice with and without DMBA, as well as their respective controls, had comparable rates of tumor-free survival. (**B**) Representative images of hematoxylin-and-eosin-stained and IHC-stained anal samples for E6 and E7 *K14E6/E7* from mice after completing the 20-week treatment period. All scale bars equal 100 μm. (**C**) Mean E6 and E7 protein expression based on IHC *K14E6/E7* anal tissue. Anal tissue was harvested from *K14E6/E7* transgenic mice either left untreated (N = 12), treated with topical DHA (N = 9), treated once weekly with topical DMBA (N = 12), or given topical DHA in conjunction with DMBA (N = 12). Unpaired *t*-tests were performed to compare no treatment to DHA only and DMBA only to DHA + DMBA. Data are presented as mean ± SEM (standard error of the mean). Significance was assigned as not significant (ns), *** *p* < 0.001.

**Figure 2 viruses-14-01632-f002:**
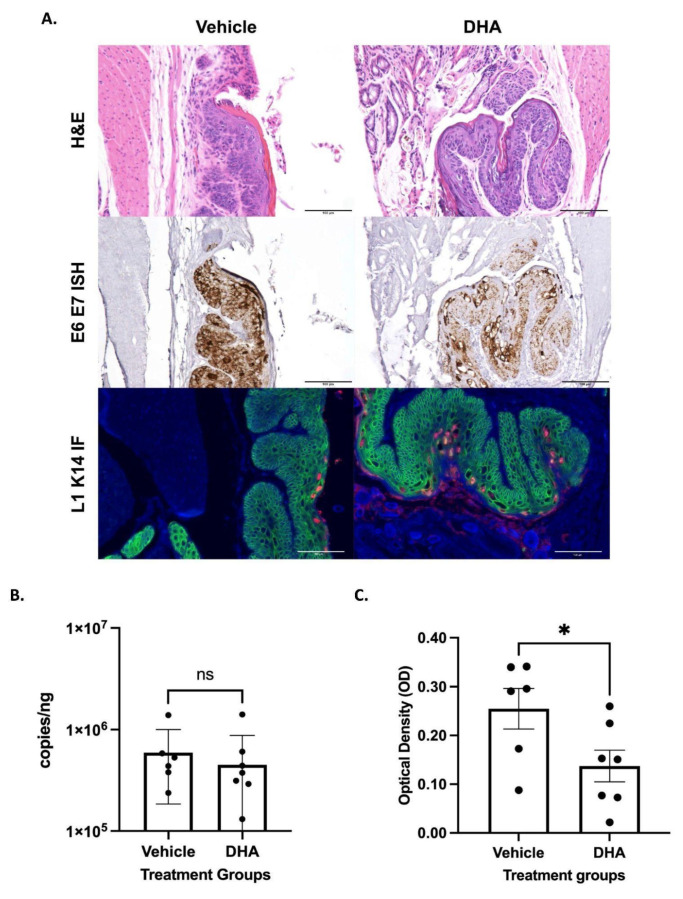
*NSG* mice infected with MmuPV1 in the anal canal began treatment 20 weeks post infection with either topical DHA (N = 8) or vehicle (N = 6). (**A**) Representative images from MmuPV1-infected NSG mouse samples after completing treatment and stained with hematoxylin and eosin, RNAscope ISH (RNAscope^®^ Probe: MusPV-E6-E7), and immunofluorescence staining for L1 (red) and K14 (green). All scale bars equal 100 μm. (**B**) Viral load analysis from *NSG* mice infected with MmuPV1 in the anus. Anal lavages were performed on vehicle mice (N = 6) and DHA-treated mice (N = 8) prior to sacrifice. DNA was extracted from the lavages, and qPCR was run for the E2 gene and normalized against 18s RNA. Treatment groups were compared via Mann–Whitney test. (**C**) RNAscope ISH anal tissue analysis from MmuPV1-infected NSG mice. Groups, vehicle mice (N = 6), and DHA-treated mice (N = 7) were compared via an unpaired *t*-test. Significance was assigned as not significant (ns), * *p* < 0.05.

**Figure 3 viruses-14-01632-f003:**
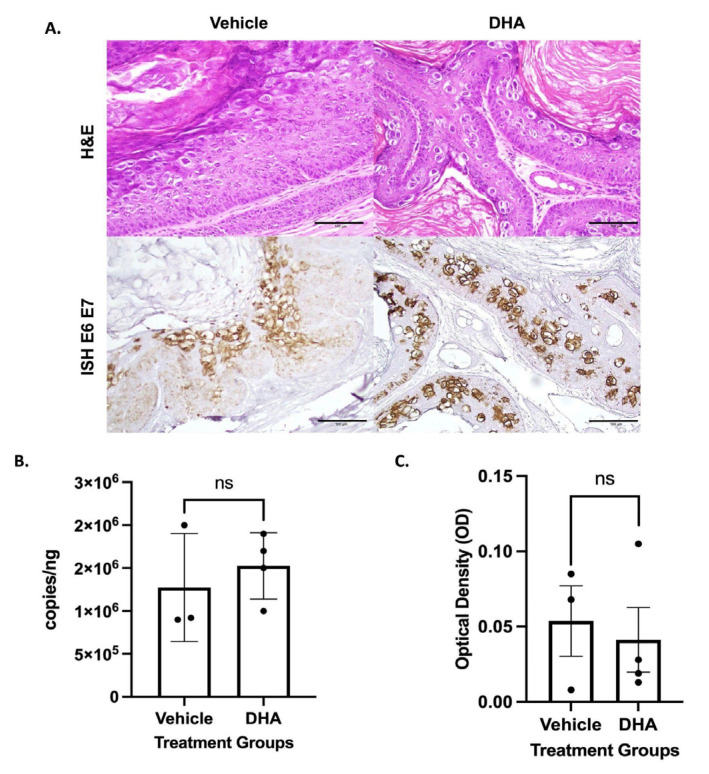
*NSG* mice infected with MmuPV1 in the cervicovaginal canal began treatment ~26.5 weeks post infection with either topical DHA (N = 4) or vehicle (N = 3). (**A**) Representative images from MmuPV1-infected NSG vaginal and vulvar tissue stained with hematoxylin and eosin and RNAscope ISH (RNAscope^®^ Probe: MusPV-E6-E7). All scale bars equal 100 μm. (**B**) Viral load analysis from *NSG* mice infected with MmuPV1. Vaginal lavages were performed on vehicle mice (N = 3) and DHA-treated mice (N = 4) prior to sacrifice. DNA was extracted from the lavages, and qPCR was run for the E2 gene and normalized against 18s RNA. Treatment groups were compared via Mann–Whitney test. (**C**) RNAscope ISH vaginal tissue analysis from MmuPV1-infected *NSG* mice. Mice were sacrificed, and tissue from the uterine horns to the vulva was harvested. Tissue was stained with RNAscope^®^ Probe: MusPV-E6-E7. Groups, vehicle mice (N =3), and DHA-treated mice (N = 4) were compared via an unpaired *t*-test. Significance was assigned as not significant (ns).

**Table 1 viruses-14-01632-t001:** Anal disease incidence in *K14E6/E7* transgenic mice. Chi-square analysis and Fisher’s exact tests were performed, comparing no treatment to DHA only and DMBA only to DHA + DMBA. There were no statistically significant differences in final anal histology between treatment groups.

Group	Non-Dysplastic	Low-Grade Dysplasia	High-Grade Dysplasia	Squamous Cell Carcinoma	Total
No treatment	2	5	5	0	12
DHA only	0	4	5	0	9
DMBA only	0	0	3	9	12
DHA + DMBA	0	0	1	11	12

**Table 2 viruses-14-01632-t002:** Final anal histology after the treatment period in MmuPV1 anally infected mice. All mice had high-grade anal dysplasia upon final pathology.

Group	Non-Dysplastic	Low-Grade Dysplasia	High-Grade Dysplasia	Squamous Cell Carcinoma	Total
Vehicle	0	0	6	0	6
DHA	0	0	8	0	8

**Table 3 viruses-14-01632-t003:** Final histology of the cervicovaginal tract after MmuPV1 infection following the treatment period. All mice developed low-grade dysplasia.

Group	Non-Dysplastic	Low-Grade Dysplasia	High-Grade Dysplasia	Squamous Cell Carcinoma	Total
Vehicle	0	3	0	0	3
DHA	0	4	0	0	4

## Data Availability

All relevant data are provided within the article.

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
