# Peer review of "Efficacy of Topically Administered Dihydroartemisinin in Treating Papillomavirus-Induced Anogenital Dysplasia in Preclinical Mouse Models"

_viruses, 2022, doi:10.3390/v14081632_

Round 1

Author Response

Reviewer 1

  1.    Lack of response to DHA: for example, the experiments would not be powered to detect differences <50% in the treated and control groups. Also, the potential benefit of DHA as part of combinatorial strategy is not assessed.

Thank you for your comment, these limitations have been added to the discussion.

2).     Regarding the MmuPV1 model of cervical neoplasia mice were euthanized before the formation of high-grade dysplasia or SCC due to the development of dyspnea. Was this determined to be a result of the viral infection? Can the authors provide some more information on the disparities here and previous work with this model (Spurgeon et.al 2019)?

Thank you for your comment, these complications were related to the older age of the mice at the time of infection compared to the mice in the Spurgeon et al study.

Minor issues:

  1. Spurgeon is misspelled “Spurgoen” p11/Discussion

Thank you for identifying this error. It has been corrected.

  1. P4, section 2.8 “permeabilized with 2N HCl” should read “DNA denatured with 2N HCl” 

Thank you for identifying this error. It has been corrected.

Reviewer 2 Report

This manuscript uses mouse models of papillomavirus-driven carcinogenesis to assess a potential therapeutic and/or chemopreventive benefit for the use of dihydroartemisin (DHA) in HPV-related cancers. Specifically the authors employ previously characterized: transgenic (K14E6/K14E7) anal carcinogenesis mouse model and murine papillomavirus infection model  for anal and cervical neoplasia. DHA is well-tolerated,  and suitable for topical application. It was previously shown to provide treatment benefits in studies involving human cells, human subjects (preliminary results), and a canine oral papillomavirus model. Thus the study rationale is sound and the results of this study are important to publish. 

In summary, the following conclusions are well supported by the data provided:

a)     In a transgenic mouse model of anal carcinogenesis, DHA does not prevent neoplastic progression, nor does it contribute to regression of anal dysplasia. Consistent with these findings the levels of E6 and E7, which are key to the onset and maintenance of HPV-driven neoplasia were not altered by DHA treatment. (Figure 1, Table 1)

b)     In a mouse infection model of anal dysplasia DHA does not prevent the onset or severity of dysplasia (Table 2, Figure 2A). Nevertheless, a statistically significant reduction in viral transcripts was seen in the DHA treated group (Figure 2B)

c)     In a mouse infection model of cervical dysplasia DHA does not prevent the onset or severity of dysplasia /9 (Figure 3, Table 3)

While the study design is thorough, conclusions well-supported, and results are important to publish, this reviewer would like to see the authors expand their discussion of 2 key topics:

1)     Lack of response to DHA: for example, the experiments would not be powered to detect differences <50% in the treated and control groups. Also, the potential benefit of DHA as part of combinatorial strategy is not assessed.

2)     Regarding the MmuPV1 model of cervical neoplasia mice were euthanized before the formation of high-grade dysplasia or SCC due to the development of dyspnea. Was this determined to be a result of the viral infection? Can the authors provide some more information on the disparities here and previous work with this model (Spurgeon et.al 2019)?

Minor issues:

Spurgeon is misspelled “Spurgoen” p11/Discussion

P4, section 2.8 “permeabilized with 2N HCl” should read “DNA denatured with 2N HCl” (Sasaki K, Adachi S, Yamamoto T, Murakami T, Tanaka K, Takahashi M. Effects of denaturation with HCl on the immunological staining of bromodeoxyuridine incorporated into DNA. Cytometry. 1988 Jan;9(1):93-6. doi: 10.1002/cyto.990090115. PMID: 2457476.)

Author Response

Reviewer 2

  1.  Page 7. The data in Fig 1C are interpreted as “There was a statistically significant greater mean E6 protein expression in the DHA + DMBA group as compared to the DMBA only group (p-value = 0.0003).” However it may be more accurate to state that “There was a statistically significant reduction in mean E6 protein expression…..” given that the DHA+DMBA E6 levels were similar to untreated controls. These differences were not observed in the E7 expression levels. Any comments as to why the DHA-treated E6 levels appeared to be lower than all three other groups? 

The mouse group with decreased E6 expression was the DMBA only group.

  1. Bottom page 7. “optical density (Figure 2A).”; Should be (Figure 2C). 

Thank you for noticing our error, this has been corrected.

  1.  Figure 2C. Regarding the reduced RNA scope signal. Is this possible evidence for integration? Was the viral DNA tissue copy number different for these two groups? 

Thank you for these questions. We added to the discussion the potential of effect, but not able to determine integration based on these studies.

  1. Failure to clear/impact anal and cervicovaginal MmuPV1 infections in contrast to some success in HPV cervical treatment clinical trials with related compounds could also indicate that the latter “success” may include activation of an adaptive immune component (absent in NSG mice).

Thank you for this comment, it has been added to the discussion.